# LIGHTWEIGHT QUAD BAYER HYBRIDEVS DEMOSAIC-ING VIA STATE SPACE AUGMENTED CROSS-ATTENTION

**Anonymous authors #9614**

## ABSTRACT

Event cameras like the Hybrid Event-based Vision Sensor (HybridEVS) camera capture brightness changes as asynchronous "events" instead of frames, offering advantages over traditional cameras: high temporal resolution, wide dynamic range, and no motion blur. However, challenges arise from combining a Quad Bayer Color Filter Array (CFA) sensor with event pixels lacking color information, resulting in aliasing and artifacts on the demosaicing process before downstream application. Current methods struggle to address these issues, especially on resource-limited mobile devices. In response, we introduce **TSANet**, a lightweight **T**wo-stage network via **S**tate space augmented cross-**A**ttention, which can handle event pixels inpainting and Quad Bayer demosaicing separately, leveraging the benefits of dividing complex tasks into manageable subtasks and learning them through a two-step training strategy to enhance robustness. Additionally, we propose a lightweight Cross-Swin State Block (CSSB) designed to augment the model's capacity to capture global dependencies using state space models in a linear format, along with cross-modality Swin attention to integrate additional priors like CFA pattern and event map, outperforming traditional local attention mechanisms while also reducing model size. In summary, TSANet demonstrates excellent demosaicing performance on HybridEVS while maintaining a lightweight model, averaging better results than the previous state-of-the-art method DemosaicFormer across seven diverse datasets in both PSNR and SSIM, while respectively reducing parameter and computation costs by $1.86\times$ and $3.29\times$. Our approach presents new possibilities for efficient image demosaicing on mobile devices. *Code and models are available in supplementary materials.*

## 1 INTRODUCTION

In recent years, event cameras have made significant progress as a new type of image sensor. Compared to conventional digital cameras, event cameras can capture event information by sensing the intensity change of specialized event pixels, thereby capturing information about objects' movement Litzenberger et al. (2006); Lichtsteiner et al. (2008). This additional visual data enables the precise capture of fast-moving objects, presenting extensive potential across various domains including robotics, automotive technology, and drone systems. However, basic imaging before downstream applications is essential for event cameras, which has not received adequate attention. On the other hand, with the development of mobile photography technology, it has been found

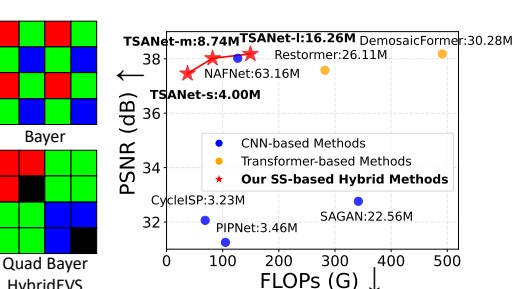

Figure 1: Left: Bayer CFA vs. Quad Bayer CFA on HybridEVS. Right: PSNR vs. FLOPs running on seven datasets, each model indicates its respective parameters on the right side. Our state space (SS) based TSANet presents three different sizes, each delivering optimal performance across various complexity ranges.

that traditional Bayer CFA sensors are constrained by their design on mobile devices, making it difficult to attain high-quality images in low-light scenes. Therefore, non-Bayer CFA sensors have gradually become mainstream in mobile photography in recent years, utilizing specifically designed

CFA to enhance low-light imaging performance. Quad Bayer CFA is one of the most popular formats, which can acquire high-resolution images on common scenes and enhance low-light imaging by pixel-binning Yoo et al. (2015). However, different from Bayer CFA which has been proposed over several decades, exploration of Quad Bayer CFA is still very limited Yang et al. (2022).

A type of event camera design named Hybrid Event Vision Sensors (HybridEVS) Kodama et al. (2023) involves utilizing a Quad Bayer CFA on a camera sensor and allocating certain pixels as event pixels to capture motion data instead of RGB color information, as shown in Fig. 1. This design introduces the camera with improved low-light imaging capabilities and the ability to capture high-speed objects effectively. However, the non-conventional Quad Bayer arrangement and the absence of color information at event pixel locations pose challenges for the demosaicing process, traditional methods face difficulties in extracting patterns from such complex arrangements, resulting in reduced imaging quality and poor performance in downstream applications such as deblurring and object detection.

Specifically, several challenges for demosaicing in Quad Bayer HybridEVS cameras are encountered: i) How to alleviate the decrease in reconstruction quality resulting from the absence of color information at event pixels locations; ii) How to realize Quad Bayer demosaicing joint with denoising; iii) How to reduce the model's parameter to make it valuable for practical applications in edge computing. Recent studies Zhou et al. (2018); Kim & Kim (2019) have proposed some end-to-end demosaicing methods based on convolution neural networks (CNNs). However, these methods fail to produce the desired results when directly applied to Non-Bayer images. Some methods have been proposed to tackle the problem of joint demosaicing and denoising of Quad Bayer CFA. A Sharif et al. (2021) proposed a method combining generative adversarial networks, utilizing depth and spatial attention mechanisms and perceptual loss to improve the reconstruction quality of mosaic images. Zeng *et al.* Zeng et al. (2023) proposed a dual-head network to transform noisy Quad Bayer into noise-free Bayer for demosaicing tasks. However, these methods do not take into account the presence of event pixels, causing color distortion and artifacts. At the same time, these networks are mostly limited by their large parameters and computational complexity, making them difficult to apply to mobile devices.

To address the above issues, we introduce TSANet, a novel lightweight two-stage model that effectively combines the position information and color information of pixel arrangements, augmented by the state space model to further explore long-range relationships inside HybridEVS RAW images. Specifically, to improve computational efficiency while enhancing reconstruction quality, we divide the complex task of joint demosaicing and denoising for HybridEVS into two stages as shown in Fig. 2. The initial stage, termed Quad-to-Quad (Q2Q), is dedicated to inpainting event pixels and denoising. Subsequently, the second stage is tailored for Quad Bayer demosaicing, called Quad-to-RGB (Q2R). Both stages employ U-Net like networks, with extra position branch for each sub-network. Considering the computational efficiency and task complexity, we utilize a network with fewer parameters in the first stage and a network with a larger parameters in the second stage. A two-step training strategy is additionally employed to effectively increase the stability and robustness.

For the design of specific models, inspired by Zheng et al. (2024); Sun et al. (2022), we use two branches in the encoders of both sub-network, extend a extra position branch to utilize position information compared to common U-Net structure. This design introduces an explicit position encoding, enabling the network to have prior knowledge of positional relationships. Specifically, for the fusion in Q2R stage, we introduce a Cross-Swin State Block (CSSB) (See Sec. 3.2), which contains Quad Bayer Cross Swin Attention (QCSA) for local cross-modality attention within windows and the Residual Vision State Space (RVSS) Zhu et al. (2024) Model in parallel for effective long-range representation capture. The hybrid design fuses local window attention across position and image, while efficiently incorporating global spatial information through RVSS with linear complexity. The tailored two-branch block ensures a balance between local and global information, optimizes both performance and efficiency. A variant of CSSB named Conv State Block (CSB) is designed for Q2R decoder, focus on inner local-global relationship of the image. We also customized a simplified dot-product-based attention called Spatial Position Attention (SPA)(See Sec. 3.3) to integrate Event map and Quad Bayer pattern while significantly reducing computational load.

In summary, our contributions are as follows:

- We propose TSANet, a lightweight Two-stage network via State space augmented cross-Attention designed for Quad Bayer HybridEVS Demosaicing. By employing a designed sub-tasks allocation and dual-branch encoders, TSANet achieves state-of-the-art performance with fewer computational resources (See Fig. 1).

- We present two unique state space augmented cross-attention blocks, termed Cross-Swin State Block (CSSB) and Conv State Block (CSB). The combined use of the Residual Vision State Space (RVSS) module with local attention and convolution demonstrates great effectiveness advantages, leading to outstanding capabilities in local-global feature extraction in a linear format.

- We design two cross-modality attention mechanisms between position and image information, named Quad Bayer Cross Swin Attention (QCSA) and Spatial Position Attention (SPA). The two modules, based on local window attention and dot product, respectively, effectively capture position features, facilitating information exchange and integration across different modalities.

## 2 RELATED WORK

**Bayer Demosaicing**   Traditional approaches for Bayer demosaicing primarily rely on interpolation techniques Hirakawa & Parks (2006), utilizing methods like adaptive algorithm Hirakawa & Parks (2005) and spatial-spectral correlations Li et al. (2008) to reconstruct full-color images. Recently, the success of convolutional networks (CNNs) used in deep learning has led to great progress in demosaicing Syu et al. (2018); Tan et al. (2017b;a; 2018); Liu et al. (2020). These methods replace traditional interpolation techniques with deep neural networks, leveraging the powerful fitting capabilities of neural networks to achieve excellent results. Some researchers Liu et al. (2020); Guo et al. (2021); Zhang et al. (2022b) proposed a demosacing network to utilize internal image information like color prior and CFA arrangement. However, most of these methods can't be extended to non-Bayer CFA formats, facing limitations like artifacts and aliasing when dealing with more challenging non-Bayer formats like Quad Bayer.

**Quad Bayer Demosaicing**   In recent years, Quad Bayer has become a popular CFA pattern widely used in mobile photography, such as smartphone cameras Yang et al. (2022). Different from traditional Bayer CFA, exploration of the Quad Bayer CFA is limited. The larger gaps between pixels of the same color make the task more challenging compared to Bayer CFA. Some two-stage networks Jia et al. (2022); Zeng et al. (2023) are proposed for progress learning to enhance Quad Bayer demosaicing. Zheng et al. (2024) proposed a dual-encoder structure to achieve better joint demosaicing and denoising tasks. GAN-based networks A Sharif et al. (2021); Sharif et al. (2021) are also used to strengthen the restoration of RGB images for Non-Bayer CFA sensors. The sensor's CFA offers a strong positional prior, yet most methods overlook the valuable color arrangement information, resulting in significant color distortion in Quad Bayer HybridEVS demosaicing.

**Event Camera Imaging**   As a novel type of sensor that has emerged in recent years Gallego et al. (2020); Son et al. (2017), the imaging research of event cameras is an active topic. Recent studies have mainly focused on imaging or downstream applications like object detection Zhang et al. (2022a) based on event information. Munda et al. (2018) try to reconstruct intensity images with direct event integration. Scheerlinck et al. (2020) introduced a fast image restoration method with deep neural networks. These methods have demonstrated excellent performance in visual tasks based on event information. However, for the latest proposed hybrid event-based vision sensor Kodama et al. (2023), the field is almost blank in advanced methods for its Image Sensor Pipelines (ISPs). A crucial step in ISPs is the demosaicing process, which converts Quad Bayer data into the RGB domain and directly impacts the imaging quality for downstream applications.

**State Space Models**   In recent years, State Space Models (SSMs) Gu et al. (2021a;b) have emerged as competitive rivals to traditional deep learning architectures like Convolution Neural Networks(CNNs) and Transformers. Pioneering works like S4 Gu et al. (2021a) and S5 Smith et al. (2022) introduced advancements on deep-state models with efficient parallel scan, modeling long-range dependency. Recently proposed Mamba Gu & Dao (2023), featuring a data-dependent SSM layer, has shown remarkable performance, surpassing Transformers in natural language tasks and demonstrating linear scalability in sequence length. Additionally, some works have applied Mamba to various vision tasks, including image classification Zhu et al. (2024), video understanding Wang et al.

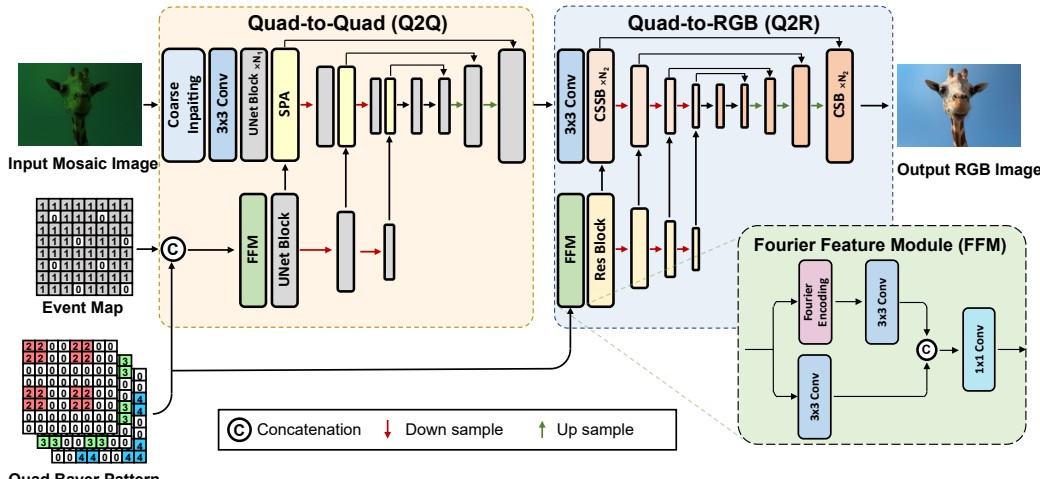

Figure 2: Overview of the TSANet approach. We adopt a two-stage structure that breaks down complex tasks into manageable subtasks, while leveraging additional branches to utilize position prior. Section 3.2 introduces the Spatial Position Attention (SPA) and Cross-Swin State Block (CSSB).

(2023), image restoration Guo et al. (2024), and image segmentation Liao et al. (2024), demonstrates the potential in visual tasks for lightweight models.

## 3 PROPOSED METHOD

Initially, we present the overall pipeline of our two-stage model in Sec. 3.1. Then we introduce the proposed state space augmented-cross attention block and its variants in Sec. 3.2. Finally, we discuss details of our proposed attention mechanisms across position and image in Sec. 3.3.

### 3.1 TWO-STAGE NETWORK STRCTURE

For the task of Quad Bayer HybridEVS demosaicing, our goal is to recover a three-channel RGB image $\mathbf{I_R} \in \mathbb{R}^{H \times W \times 3}$ from a degraded Quad Bayer image $\mathbf{I_Q} \in \mathbb{R}^{H \times W \times 1}$, where the degradation includes color channel loss $D_q$ due to the Quad Bayer CFA, pixel absence $D_e$ due to the design of event pixels in the sensor, and noise $D_n$ introduced during the image capture process, respectively. Most previous methods aim to directly learn the entire process through an all-in-one deep model $\mathcal{M}$ that restores image $\mathbf{I_R}$ from $\mathbf{I_Q}$, which can be expressed as:

$$\mathbf{I_R} = \mathcal{M}(\mathbf{I_Q}). \tag{1}$$

However, these all-in-one models often struggle to extract the inner connection between position and color, causing unbearable aliasing and artifacts (See Fig. 6), or require a large number of parameters and computation load to achieve ideal restoration results, make it barely impossible to deploy on limited-resource mobile devices. Unlike past single-model solutions, we define the composite task as two controllable sub-tasks: the former sub-task is to restore the degradation of $D_e$ and $D_n$ from the original Quad Bayer image, inpainting absent pixels and reducing noise, producing a clean Quad Bayer image, defined as $\mathcal{M}^{Q2Q}$; the later one is to restore the clean Quad Bayer image into $\mathbf{I_R}$, defined as $\mathcal{M}^{Q2R}$. The overall pipeline progressively restore the RGB image from the degraded Quad Bayer image. Notably, we introduce a distinctive encoder branch designed to integrate position information as a dedicated prior knowledge into the network, using position information $\mathbf{P_e} \in \mathbb{R}^{H \times W \times 1}$ from $D_e$ and $\mathbf{P_q} \in \mathbb{R}^{H \times W \times 3}$ from $D_q$ to achieve better image reconstruction, the process can be expressed as:

$$\mathbf{I_R} = \mathcal{M}^{Q2R}(\mathcal{M}^{Q2Q}(\mathbf{I_Q}, (\mathbf{P_q}, \mathbf{P_e})), \mathbf{P_q}). \tag{2}$$

Our proposed two-stage network architecture is depicted in Fig. 2. Such a design not only assigns specific tasks to sub-networks but also benefits from a two-step training strategy. Prior studies have demonstrated that pretraining on sub-networks can lead to improved performance and inference

stability. Our designed two-stage architecture facilitates directional pretraining for the two sub-networks by synthesizing a dummy clean Quad Bayer.

Besides, we believe fully utilizing position information is crucial for the demosaicing problem of Quad Bayer, especially in the task with event pixels. Therefore, we propose an additional positional encoding branch in both networks, explicitly integrating position information into the network. To further enhance the model's ability to extract high-frequency texture information, we designed a **F**ourier **F**eature **M**odule (FFM) based on Fourier encoding, as shown in Fig. 2, which maps the position information to a series of high-frequency features based on sine and cosine functions Tancik et al. (2020), enabling the model to capture refine details and patterns in position dimension, strengthening the restoration results of complex textures. At the Q2Q stage, specially, before putting the Quad Bayer image into the network, we apply a coarse inpainting by averaging nearby pixels around event pixels, which aims to mitigate color loss resulting from event pixels.

### 3.2 STATE SPACE AUGMENTED BLOCKS

In this section, we propose two lightweight modules augmented by State Space Models for the encoder and decoder of the Q2R stage, respectively. We begin with a dual-branch structure named Cross-Swin State Block (CSSB), concurrently modeling local cross-modality attention and long-range dependencies with linear complexity. Then, we present its variant Conv State Block (CSB) in the decoder, enhancing local feature restoration while further reducing computation by convolutions.

**Cross-Swin State Block** Fig. 3a illustrated our proposed Cross-Swin State Block (CSSB). This block is designed to capture long-range dependencies and cross-modality local attention in parallel. It integrates Residual Vision State Space (RVSS) and Quad Bayer Cross Swin Attention (QCSA) mentioned in Sec. 3.3. Fig. 4c shows RVSS, a simplified version of Residual State Space Block of MambaIR Guo et al. (2024), preserving its core component VSSM for efficient extraction of long-range dependencies, followed by a residual connection. To further reduce computational complexity while capturing local positional attention intersections and global long-range dependencies simultaneously, we follow SCUNet Zhang et al. (2023) and propose a parallel network module. First, the image feature projected through 1x1 convolution is split and separately inputted into QCSA and RVSS modules. The outputs are then concatenated and utilized for out projection through a $1 \times 1$ convolution, which is followed by a residual connection. For a image input $\mathbf{F_I}$ and a position input $\mathbf{F_P}$, the process can be expressed as follows:

$$X_1, X_2 = \text{Split}(\text{Conv}_{1\times 1}(\mathbf{F_I})),$$
$$Y_1, Y_2 = \text{QCSA}(X_1, \mathbf{F_P}), \ \text{RVSS}(X_2), \tag{3}$$
$$\hat{\mathbf{F}}_\mathbf{I} = \text{Conv}_{1\times 1}(\text{Concat}(Y_1, Y_2)) + \mathbf{F_I}.$$

The QCSA primarily extracts representations with position information, models spatial information through the two modalities of position and image while RVSS is used to effectively capture global information, which parallel address cross-modality local attention and long-range dependencies. Moreover, the dual-brunch structure is similar to group convolution, reducing the number of channels within the module through splitting operations, effectively reducing the computational complexity and parameters of the block.

**Conv State Block** We also propose a variant of CSSB for the decoder of the Q2R stage, as shown in Fig. 3b. Instead of employing QCSA to capture attention features across position information, we replace this module with Residual Convolution (RConv) Zhang et al. (2021), which focuses on restoring internal local feature of the image, parallel with an RVSS to enhance long-range feature extraction capability, thereby forming a lightweight decoder block with local-global dependencies.

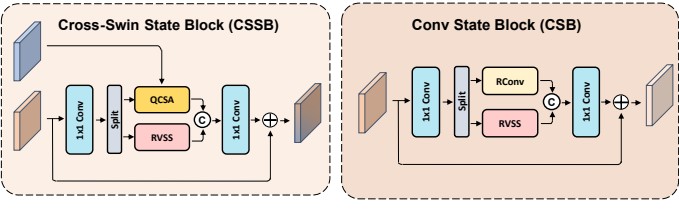

(a) Cross-Swin State Block          (b) Conv State Block

Figure 3: Proposed state space augmented blocks. The two blocks are modified from Transformer and Convolution for the Q2R encoder and decoder, respectively. The dual-branch design parallel extracts feature with local-global dependencies while reducing computation load.

(a) Quad Bayer Cross Swin Attention     (b) Spatial Position Attention     (c) Residual Vision State Space

Figure 4: Illustration of designed modules. We propose two cross-attention modules and one state space model. Attentions focus on fusing position information into network and RVSS captures long-range dependencies.

## 3.3 ATTENTION MODULES FOR POSITION FUSION

As illustrated in Fig. 4a and Fig. 4b, we propose two spatial attention modules designed for fusing position information and image feature. We utilize the position representations as extra information and explore spatial relationship of position and image.

**Quad Bayer Cross Swin Attention** As shown in Fig. 4a, we designed a sufficient feature fusion mechanism for Q2R sub-network. It first employs an efficient Window Multi-head Cross Attention (WMCA) inspired by Swin Transformer Liu et al. (2021). Specifically, given image feature $\mathbf{F_I} \in \mathbb{R}^{H \times W \times d}$ and position feature $\mathbf{F_P} \in \mathbb{R}^{H \times W \times d}$, both features are first partitioned into non-overlapping $M \times M$ local windows, getting $\frac{HW}{M^2} \times M^2 \times d$ features. Different from conventional attention acquiring all $query(Q)$, $key(K)$, $value(V)$ from $\mathbf{F_I}$, we produce $Q$ from $\mathbf{F_P}$. The $Q$, $K$ and $V$ for window feature $X \in \mathbb{R}^{M^2 \times d}$ from $\mathbf{F_I}$ and $Y \in \mathbb{R}^{M^2 \times d}$ from $\mathbf{F_P}$ are computed as:

$$Q = YP_Q, \quad K = XP_K, \quad V = XP_V. \tag{4}$$

where $P_Q$, $P_K$ and $P_V$ are shared project matrices among each window. Then we have $Q, K, V \in \mathbb{R}^{M^2 \times d}$ to compute in-window cross attention as:

$$\text{Attention}(Q, K, V) = \text{SoftMax}\left(\frac{QK^T}{\sqrt{d}} + B\right). \tag{5}$$

where $B$ represents relative positional encoding within the window, complementing the global positional encoding introduced by $\mathbf{F_P}$. This attention mechanism operates across all windows and is executed in parallel $h$ times Vaswani et al. (2017). After WMCA, the features are then fed into a two-layer multi-layer perceptron (MLP) with GELU Hendrycks & Gimpel (2016) activation function for further feature extraction. Both steps utilize residual connections and LayerNorm Ba et al. (2016), the process can be expressed as:

$$\begin{aligned}
\hat{\mathbf{F}}_\mathbf{I} &= \text{WMCA}(\text{LN}(\mathbf{F_I}), \text{LN}(\mathbf{F_P})) + \mathbf{F_I}, \\
\hat{\mathbf{F}}_\mathbf{I} &= \text{MLP}(\hat{\mathbf{F}}_\mathbf{I}) + \hat{\mathbf{F}}_\mathbf{I}.
\end{aligned} \tag{6}$$

Then, through the shifted window mechanism Liu et al. (2021), this module achieves cross-window information exchange. It incorporates global information of Quad Bayer CFA pattern into the network, offering a spatial attention mechanism based on position information. Additionally, the window mechanism maintains the computational complexity of the network linearly, effectively controlling computational and parameter overhead compared to conventional attention methods.

**Spatial Position Attention** As shown in Fig. 4b, we designed another cross-modality attention mechanism, aimed at building the relationship between position and image sufficiently in the Q2Q stage. Specifically, the position branch introduces an explicit representation of the spatial dimension. Firstly, the position feature $\mathbf{F_P} \in \mathbb{R}^{H \times W \times d}$ and the image feature $\mathbf{F_I} \in \mathbb{R}^{H \times W \times d}$ pass through their respective convolutional projection layers. Subsequently, the position branch is activated by ReLU, after which they demonstrate element-wise product with each other. The process can be expressed as:

$$\hat{\mathbf{F}}_\mathbf{I} = \text{Conv}_{1 \times 1}\left(\text{LN}\left(\mathbf{F_I}\right)\right) \odot \text{ReLU}(\text{Conv}_{1 \times 1}(\text{LN}\left(\mathbf{F_P}\right)) + \mathbf{F_I}. \tag{7}$$

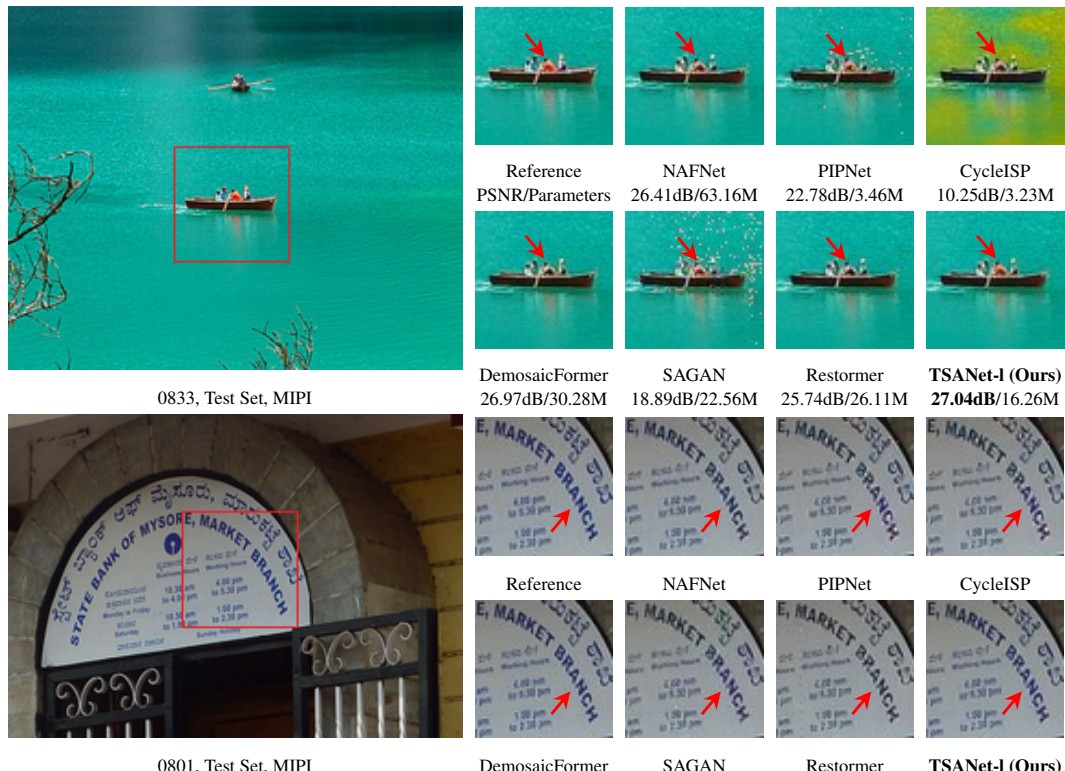

Figure 5: Visualized results of all compared methods for Quad Bayer HybridEVS Demosaicing on MIPI dataset. The proposed TSANet demonstrates the best visual results among all methods, producing more vivid colors on scenes with complex coloration, better than the previous state-of-the-art approach Restormer.

The position information is calculated to a weights map of image feature, determining the importance of each feature pixel. It is similar to a Gate Mechanism Zamir et al. (2022), which can control the flow of information but by position information rather than itself. Additionally, this method exhibits faster computational performance when compared to convolution or attention mechanisms.

## 4 EXPERIMENTS

In this section, we first introduce experimental settings including implementation details and datasets. Then we compared our proposed TSANet with other state-of-the-art methods on seven diverse datasets. Finally, we demonstrate an ablation study to prove the effectiveness of our methods.

### 4.1 EXPERIMENTAL SETTINGS

**Implementation details**   In all experiments, we use the following hyperparameters, unless mentioned otherwise. During the training, we randomly crop the original Quad Bayer input and RGB ground truth into $128 \times 128$ patches, with batch size = 32. We use Adam as the optimizer and the learning rate starts from $2 \times 10^{-4}$ and is gradually reduced to $1 \times 10^{-7}$ with the cosine annealing scheme. The total iteration is set to $1 \times 10^{6}$. Specifically, to fully utilize the two-stage structure of our TSANet, we apply a pretraining step on the sub-network before end-to-end joint training. In particular, we employ published pretrained weights of DemosaicFormer for testing.

**Datasets**   We train all the models with the dataset from Mobile Intelligent Photography & Imaging (MIPI) Workshop 2024 Demosaic for Hybridevs Camera challenge trackWu et al. (2024), which contains 800 pairs of Quad Bayer and RGB images with 2K resolution. The official public test set of MIPI dataset contains 26 pairs. Both the training and testing data includes real world noise. we simulate HybridEVS pattern test cases from five image datasets, including Kodak Loui et al. (2007), Urban100 Cordts et al. (2016), BSD100 Martin et al. (2001), and Wed Ma et al. (2017) (first 100 images). Additionally, we assess dynamic performance using two video datasets: REDS Nah et al.

| Methods | Params (M) | FLOPs (G) | Image Datasets | | | | | Video Datasets | | Average |
|---|---|---|---|---|---|---|---|---|---|---|
| | | | Kodak | BSD100 | Urban100 | Wed | MIPI | REDS | Vid4 | |
| | | | PSNR/SSIM | | | | | | | |
| DemosaicFormer | 30.28 | 491.1 | 39.32/0.982 | **37.65**/0.982 | **37.64**/0.980 | 34.86/0.968 | **39.35/0.981** | 42.45/0.991 | **36.01**/0.979 | 38.18/0.980 |
| **TSANet-l (Ours)** | 16.26 | 149.4 | **39.40/0.986** | 37.34/**0.986** | 37.07/**0.983** | 35.76/0.977 | 39.07/0.980 | **43.00/0.996** | 35.64/**0.982** | 38.18/**0.984** |
| NAFNet | 63.16 | 126.7 | 39.14/0.985 | **37.51**/0.986 | 36.64/0.982 | **35.73**/0.969 | 38.89/0.979 | 42.76/0.996 | 35.48/0.981 | 38.02/0.983 |
| Restormer | 26.11 | 282.2 | 39.16/0.986 | 37.11/0.985 | 36.36/0.977 | 35.00/0.971 | 38.42/0.978 | 41.91/0.990 | 35.08/0.980 | 37.58/0.981 |
| SAGAN | 22.56 | 341.6 | 36.14/0.974 | 30.53/0.931 | 29.89/0.946 | 28.22/0.917 | 34.25/0.959 | 38.13/0.984 | 32.16/0.963 | 32.76/0.953 |
| **TSANet-m (Ours)** | 8.74 | 81.94 | 39.24/**0.986** | 37.25/**0.986** | 36.75/**0.982** | 35.60/**0.976** | **38.93**/0.979 | 42.87/**0.996** | **35.53**/**0.982** | **38.02**/**0.984** |
| PIPNet | 3.46 | 68.8 | 32.20/0.960 | 31.97/0.950 | 28.92/0.942 | 29.19/0.929 | 33.73/0.950 | 36.19/0.981 | 32.20/0.964 | 32.06/0.954 |
| CycleISP | **3.23** | 104.9 | 33.09/0.970 | 32.18/0.969 | 29.78/0.942 | 30.22/0.944 | 30.04/0.934 | 32.96/0.975 | 30.46/0.964 | 31.25/0.957 |
| **TSANet-s (Ours)** | 4.00 | 37.4 | **38.73/0.984** | **36.56/0.984** | **36.15/0.980** | **35.19/0.973** | **38.47/0.978** | **41.94/0.989** | **35.15/0.980** | **37.46/0.981** |

Table 1: Quantitative evaluation of TSANet compared to other methods across seven diverse datasets. Methods with an orange background are Transformer-based, while those with a blue background are Convolution-based. Our TSANet provides three model sizes and consistently achieves the best average PSNR and SSIM across various computational complexity levels, while significantly reducing both parameters and FLOPs. Specifically, Our TSANet-l achieves the best average PSNR and SSIM scores across seven datasets, surpassing DemosaicFormer with 0.004 on SSIM while reducing parameter and computation costs by $1.86\times$ and $3.29\times$.

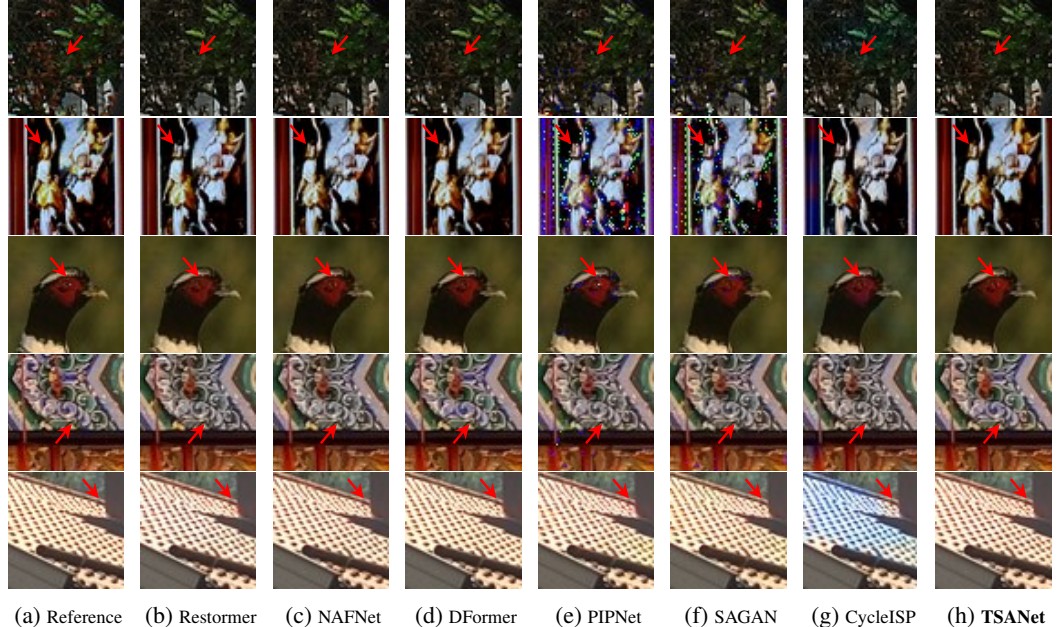

(a) Reference  (b) Restormer  (c) NAFNet  (d) DFormer  (e) PIPNet  (f) SAGAN  (g) CycleISP  (h) **TSANet**

Figure 6: Visualized results of all compared methods for Quad Bayer HybridEVS Demosaicing on synthesized image datasets. "Demosaicformer" is abbreviated as "DFormer". The comparison provides further validation of TSANet across various scenarios and confirms its effectiveness in restoring fine details, colors, and textures.

(2019) and Vid4 Liu & Sun (2011). We re-sample Quad Bayer image from RGB images and simulate event pixels to synthesize the input.

## 4.2 COMPARISON TO STATE-OF-THE-ARTS

**Quantitative Comparison.** We compare our proposed TSANet with several state-of-the-art methods, including two joint demosaicing and denoising methods PIPNet A Sharif et al. (2021) and SAGAN Sharif et al. (2021), four image restoration methods, Restormer Zamir et al. (2022) and DemosaicFormer Xu et al. (2024) based on Transformer, NAFNet Chen et al. (2022) and CycleISP Zamir et al. (2020) based on Convolution. We choose PSNR/SSIM scores as restoration quality metrics, parameters and FLOPs as computational complexity metrics, to illustrate models' performance. Table

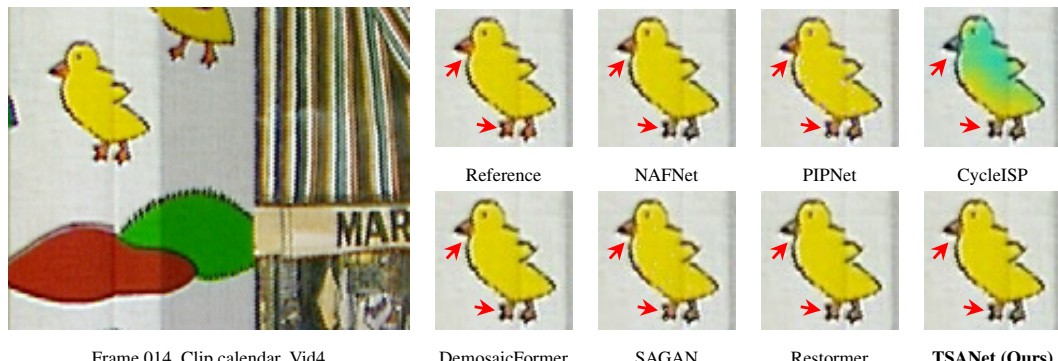

Figure 7: Visualized results of various methods for Quad Bayer HybridEVS Camera demosaicing on video datasets. The proposed TSANet exhibits finer details, showing its advantages in high-speed and dynamic scenes essential for event cameras.

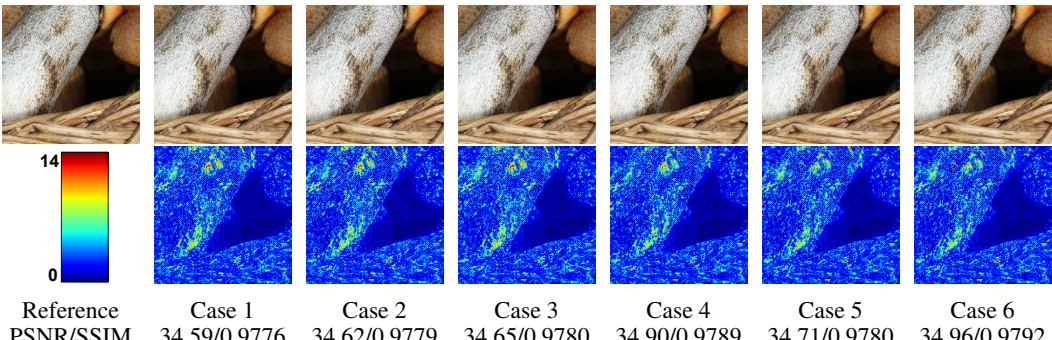

Figure 8: Visualization of ablation study. We displayed the visual comparison results with a difference map of our ablation studies, which separately validate various components, thus proving the efficacy of our module design.

1 shows the quantitative comparison results on all test datasets. It is worth noting that due to its large model size, Restormer and DemosaicFormer couldn't be tested on the 2k MIPI dataset, so we partitioned input images into 700x700 patches for inference and recombined the results. In particular, the proposed TSANet achieves state-of-the-art performance in different complexity levels across seven diverse test datasets, surpassing the previous SOTA method DemosaicFormer by 0.004 in SSIM, while reducing parameters and computations by 1.86x and 3.29x respectively. Additionally, on seven synthetic datasets, TSANet-l achieves the best PSNR/SSIM results in three and second-best results in four. Furthermore, smaller versions of TSANet-s and TSANet-m also achieve the best results on corresponding complexity ranges. Experimental comparisons validate the effectiveness of our approach, ensuring restoration performance while drastically reducing computational resources, introducing a model friendly to edge devices with limited computational resources.

**Visualization Comparison.** Fig. 5 and Fig. 6 respectively depict the visual comparison results of TSANet compared with other models on the MIPI dataset and other synthesized image datasets. Benefiting from our local-global feature extraction structure, our model exhibits closer visual similarity to ground truth in image details, particularly in subtle color variations, resulting in more vivid colors compared to other methods. While others suffer from color loss due to the Quad Bayer pattern and event pixels, leading to further degradation of color information in fine textures and even causing severe pixel errors (see Fig. 5). Further comparison on video datasets is demonstrated in Fig.7, the finer details of our TSANet demonstrate strong competitiveness in handling high-speed dynamic range scenes captured by event cameras. Overall, the perceptual visual results confirm the effectiveness of our approach.

### 4.3 ABLATION STUDY

As shown in Table 2, we present ablation experiments to validate the effectiveness of the proposed QCSA, SPA, RVSS modules and pretraining strategy in TSANet. Models are evaluated on the MIPI

dataset. First, we remove both the QCSA and SPA cross-attention modules, then add each module separately to assess their impact on network performance. As shown in Fig. 9, their combined inclusion improves PSNR by 0.06dB with only an 8% increase in parameters. Additionally, we evaluate the improvements introduced by RVSS and the two-step training strategy independently. The pretraining step can effectively improve model performance by 0.13dB. RVSS reduces parameters by 38% with just a 0.02dB drop in PSNR, highlighting the effectiveness of incorporating state space models. Visual comparison is demonstrated in Fig. 8. Results showed that attention modules effectively enhance model performance without notably increasing the number of parameters or computational load. Despite the introduction of RVSS leading to a slight performance decrease, the reduction in parameters and computational load is significant. Our final model achieves a balance between computational resources and effectiveness.

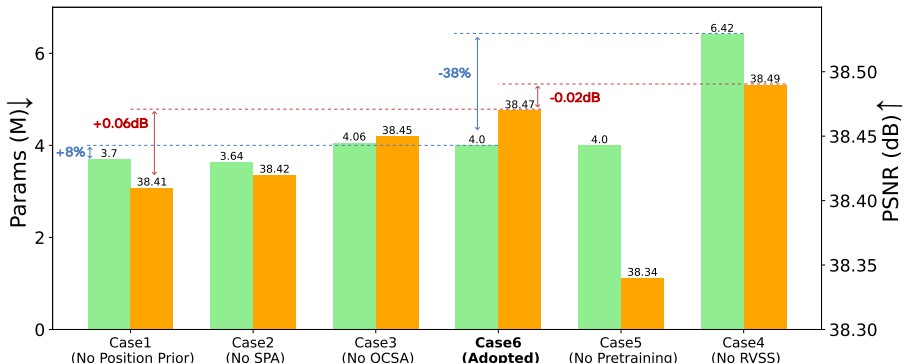

Figure 9: Quantitative visualization of the ablation study. The position prior branch we adopted significantly improves performance in PSNR by 0.06dB with only an 8% increase in parameters, while the state space modules reduce parameters by 38%, resulting in just a 0.02dB drop in PSNR.

Table 2: Quantitative results of ablation study. We validated the impact of the QCSA, SPA, RVSS modules and the two-step training strategy on TSANet-s, demonstrating the effectiveness of our proposed cross-attentions and training recipe while validating that employing RVSS can significantly reduce computational resources and maintain performance.

| Case | Modules/Strategy | | | | PARAMs (M) | FLOPs (G) | MIPI PSNR/SSIM |
|---|---|---|---|---|---|---|---|
| | QCSA | SPA | RVSS | Two-step Training | | | |
| 1 | | | ✓ | ✓ | 3.70 | 32.0 | 38.41/0.9773 |
| 2 | ✓ | | ✓ | ✓ | 3.64 | 32.5 | 38.42/0.9773 |
| 3 | | ✓ | ✓ | ✓ | 4.06 | 36.9 | 38.45/0.9775 |
| 4 | ✓ | ✓ | | ✓ | 6.42 | 59.1 | 38.49/0.9776 |
| 5 | ✓ | ✓ | ✓ | | 4.00 | 37.4 | 38.34/0.9772 |
| 6 | ✓ | ✓ | ✓ | ✓ | 4.00 | 37.4 | 38.47/0.9775 |

## 5 CONCLUSION

We have presented a novel lightweight two-stage structure network tailored for the HybridEVS architecture, which introduces task-specific sub-networks and a corresponding two-step training strategy. Specifically, we introduce Cross-Swin State Block (CSSB) and Conv State Block (CSB) strengthened by Residual Vision State Space (RVSS) to maintain low computational complexity while simultaneously addressing local position features and long-range dependencies. We further propose the Quad Bayer Cross Swin Attention (QCSA) and Spatial Position Attention (SPA) mechanisms to effectively couple the arrangement information of Quad Bayer pattern and event points in the network, providing explicit prior encoding for global position information. To the best of our knowledge, this is the first work employing SSMs in a hybrid model for demosaicing tasks. Our approach is validated across multiple datasets, offering a lightweight and mobile-friendly model for Quad Bayer HybridEVS Demosaicing.

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

# A   MODEL DETAILS IN PYTORCH STYLE PSEUDO-CODE

In this section, we provide a detailed explanation of the modules we designed, including Cross-Swin State Block, Conv State Block, Quad Bayer Cross Swin Attention and Spatial Position Attention using PyTorch style pseudo-code. **The specific runnable model code is provided in the supplementary material.**

---

**Algorithm 1** Pseudo-code of Cross-Swin State Block

---

```python
## Cross-Swin State Block (CSSB)
class CSSB(nn.Module):
    def __init__(self, rvss_dim, trans_dim, head_dim, window_size,\
    drop_path, type='W', input_resolution=None):
        super(CSSB, self).__init__()
        self.rvss_dim = rvss_dim
        self.trans_dim = trans_dim
        self.head_dim = head_dim
        # size of local window
        self.window_size = window_size
        # drop out
        self.drop_path = drop_path
        # type: 'W' (Window) or 'SW' (Shifted Window)
        self.type = type
        self.input_resolution = input_resolution
        assert self.type in ['W', 'SW']
        if self.input_resolution <= self.window_size:
            self.type = 'W'
        # input projection
        self.conv1_1 = nn.Conv2d(self.rvss_dim+self.trans_dim, \
        self.rvss_dim+self.trans_dim, 1, 1, 0, bias=True)
        self.conv1_2 = nn.Conv2d(self.rvss_dim+self.trans_dim, \
        self.rvss_dim+self.trans_dim, 1, 1, 0, bias=True)
        # employ Quad Bayer Cross Swin Attention
        self.trans_block = QCSA(self.trans_dim, self.trans_dim, self.head_dim, \
        self.window_size, self.drop_path, self.type, self.input_resolution)
        # Visual State Space Model
        self.rvss = VSSBlock(
                hidden_dim=self.rvss_dim,
                drop_path=0,
                norm_layer=nn.LayerNorm,
                attn_drop_rate=0,
                d_state=16,
                expand=2)

    def forward(self, image, position):
        # input projection and split
        rvss_x, trans_x = torch.split(self.conv1_1(image),\
        (self.rvss_dim, self.trans_dim), dim=1)
        # Residual Visual State Space Module
        rvss_x = self.rvss(rvss_x) + rvss_x
        # Rearrange inputs into 'b h w c'
        trans_x = Rearrange('b c h w -> b h w c')(trans_x)
        position = Rearrange('b c h w -> b h w c')(position)
        # Quad Bayer Cross Swin Attention
        trans_x = self.trans_block(trans_x, position)
        # Rearrange ouput back
        trans_x = Rearrange('b h w c -> b c h w')(trans_x)
        # concatenate and output projection
        res = self.conv1_2(torch.cat((rvss_x, trans_x), dim=1))
        # residual connection
        image = image + res
        return image
```

---

**Algorithm 2** Pseudo-code of Cross-Swin State Block

```python
## Conv State Block (CSB)
class CSB(nn.Module):
    def __init__(self, conv_dim, rvss_dim, head_dim, window_size,\
    drop_path, type='W', input_resolution=None):
        super(CSB, self).__init__()
        self.conv_dim = conv_dim
        self.rvss_dim = rvss_dim
        self.head_dim = head_dim

        # input projection
        self.conv1_1 = nn.Conv2d(self.conv_dim+self.rvss_dim, \
        self.conv_dim+self.rvss_dim, 1, 1, 0, bias=True)
        self.conv1_2 = nn.Conv2d(self.conv_dim+self.rvss_dim, \
        self.conv_dim+self.rvss_dim, 1, 1, 0, bias=True)

        # Visual State Space Model
        self.rvss = VSSBlock(
                hidden_dim=self.rvss_dim,
                drop_path=0,
                norm_layer=nn.LayerNorm,
                attn_drop_rate=0,
                d_state=16,
                expand=2)

        # Convolution Block
        self.conv_block = nn.Sequential(
                nn.Conv2d(self.conv_dim, self.conv_dim, 3, 1, 1, bias=False),
                nn.ReLU(True),
                nn.Conv2d(self.conv_dim, self.conv_dim, 3, 1, 1, bias=False)
                )

    def forward(self, x):
        # input projection and split
        conv_x, rvss_x = torch.split(self.conv1_1(x), \
        (self.conv_dim, self.rvss_dim), dim=1)
        # Residual Convolution
        conv_x = self.conv_block(conv_x) + conv_x
        # Residual Visual State Space Model
        rvss_x = self.rvss(rvss_x) + rvss_x
        # concatenate and output projection
        res = self.conv1_2(torch.cat((conv_x, rvss_x), dim=1))
        # residual connection
        x = x + res
        return x
```

**Algorithm 3** Pseudo-code of Quad Bayer Cross Swin Attention

```python
## Quad Bayer Cross Swin Attention (QCSA)
class QCSA(nn.Module):
    def __init__(self, input_dim, output_dim, head_dim, window_size,\
    drop_path, type='W', input_resolution=None):
        super(QCSA, self).__init__()
        self.input_dim = input_dim
        self.output_dim = output_dim
        # Swin type
        assert type in ['W', 'SW']
        self.type = type
        if input_resolution <= window_size:
            self.type = 'W'
        # Layer Norm
        self.ln1 = nn.LayerNorm(input_dim)
        self.ln2 = nn.LayerNorm(input_dim)
        self.ln3 = nn.LayerNorm(input_dim)

        # Window Mulit Head Cross Attention
        self.msa = WMCA(input_dim, input_dim, head_dim, window_size, self.type)

        # drop out
        self.drop_path = DropPath(drop_path) if drop_path > 0. else nn.Identity()

        # Multi Layer Perceptron
        self.mlp = nn.Sequential(
            nn.Linear(input_dim, 4 * input_dim),
            nn.GELU(),
            nn.Linear(4 * input_dim, output_dim),
        )

    def forward(self, image, position):
        # Cross Swin Layer + residual connection
        image = image + self.drop_path(self.msa(self.ln1(image), self.ln3(position)))

        # MLP for out projection
        fused = image + self.drop_path(self.mlp(self.ln2(image)))
        return fused
```

---

**Algorithm 4** Pseudo-code of Window Multi Head Cross Attention

---

```python
## Window Multi Head Cross-attention module in Quad Bayer Cross Swin Attention
class WMCA(nn.Module):
    def __init__(self, input_dim, output_dim, head_dim, window_size, type):
        super(WMCA, self).__init__()
        self.input_dim = input_dim
        self.output_dim = output_dim
        self.head_dim = head_dim
        self.n_heads = input_dim//head_dim
        # scale factor
        self.scale = self.head_dim ** -0.5
        # size of local window
        self.window_size = window_size
        # Swin type: 'W' (Window) or 'SW' (Shifted Window)
        self.type=type
        # positional encoding inside the window
        self.relative_position_params = nn.Parameter(torch.zeros((2 *\
        window_size - 1)*(2 * window_size -1), self.n_heads))
        trunc_normal_(self.relative_position_params, std=.02)
        self.relative_position_params = torch.nn.Parameter(self.relative_position_params.view(
        window_size-1, 2*window_size-1, self.n_heads).transpose(1,2).transpose(0,1))
        # input projection
        self.embedding_layer = nn.Linear(self.input_dim, 2*self.input_dim, bias=True)
        self.embedding_layer_qb = nn.Linear(self.input_dim, self.input_dim, bias=True)
        # output projection
        self.linear = nn.Linear(self.input_dim, self.output_dim)
    def forward(self, x, y):
        # x: input image tensor with shape of [b h w c];
        # y: input Quad Bayer tensor with shape of [b h w c];
        # shift window when type = 'SW'
        if self.type!='W':
            x = torch.roll(x, shifts=(-(self.window_size//2), -(self.window_size//2)), dims=(1
            y = torch.roll(y, shifts=(-(self.window_size//2), -(self.window_size//2)), dims=(1
        # patrition to windows
        x = rearrange(x, 'b (w1 p1) (w2 p2) c -> b w1 w2 p1 p2 c',\
            p1=self.window_size, p2=self.window_size)
        h_windows = x.size(1)
        w_windows = x.size(2)
        x = rearrange(x, 'b w1 w2 p1 p2 c -> b (w1 w2) (p1 p2) c',\
            p1=self.window_size, p2=self.window_size)
        y = rearrange(y, 'b (w1 p1) (w2 p2) c -> b (w1 w2) (p1 p2) c',\
            p1=self.window_size, p2=self.window_size)
        # input projection
        kv = self.embedding_layer(x)
        k, v = rearrange(kv, 'b nw np (threeh c) -> threeh b nw np c',\
            c=self.head_dim).chunk(2, dim=0)
        q = self.embedding_layer_qb(y)
        q = rearrange(q, 'b nw np (h c) -> h b nw np c', c=self.head_dim)
        # cross window attention
        sim = torch.einsum('hbwpc,hbwqc->hbwpq', q, k) * self.scale
        sim = sim + rearrange(self.relative_embedding(), 'h p q -> h 1 1 p q')
        # masks when shifted window
        if self.type != 'W':
            attn_mask = generate_mask(h_windows, w_windows,\
                self.window_size, shift=self.window_size//2)
            sim = sim.masked_fill_(attn_mask, float("-inf"))
        # attention map
        probs = nn.functional.softmax(sim, dim=-1)
        # attention to value
        output = torch.einsum('hbwij,hbwjc->hbwic', probs, v)
        output = rearrange(output, 'h b w p c -> b w p (h c)')
        # output projection
        output = self.linear(output)
        output = rearrange(output, 'b (w1 w2) (p1 p2) c -> b (w1 p1) (w2 p2) c',\
            w1=h_windows, p1=self.window_size)

        # shift back when type = 'SW'
        if self.type!='W': output = torch.roll(output, shifts=\
            (self.window_size//2, self.window_size//2), dims=(1,2))
        return output
```

**Algorithm 5** Pseudo-code of Spatial Position Attention

```python
## Spatial Position Attention (SPA)
class SPA(nn.Module):
    def __init__(self, dim, num_heads, bias=False, LayerNorm_type='WithBias'):
        super(SPA, self).__init__()
        # Layer Norm
        self.norm1_image = LayerNorm(dim, LayerNorm_type)
        self.norm1_position = LayerNorm(dim, LayerNorm_type)
        # Attention
        self.num_heads = num_heads
        self.temperature = nn.Parameter(torch.ones(dim, 1, 1))
        self.q = nn.Conv2d(dim, dim, kernel_size=1, bias=bias)
        self.v = nn.Conv2d(dim, dim, kernel_size=1, bias=bias)

    def forward(self, image, position):
        # image: b, c, h, w
        # position: b, c, h, w
        # return: b, c, h, w

        # projection
        q = self.q(self.norm1_image(image)) # qb
        v = self.v(self.norm1_position(position)) # image

        # position attention
        x_spatial = F.relu(q) * v * self.temperature

        # residual connection
        fused = image + x_spatial

        return fused
```

