# OpenReview forum: "Lightweight Quad Bayer HybridEVS Demosaicing via State Space Augmented Cross-Attention"
_ICLR.cc/2025/Conference — ICLR 2025 Conference Withdrawn Submission_

### Official Review · Reviewer_nXsc · 2024-10-21

**Soundness:** 2
**Presentation:** 2
**Contribution:** 2
**Rating:** 5
**Confidence:** 3

**Summary:**

This paper addresses an important issue in the domain of Hybrid Event-based Vision Sensors (HybridEVS): demosaicing for sensors that combine RGB and event pixels. Demosaicing is a fundamental problem, and solving it in the context of new sensor designs is essential. However, I have several concerns regarding the scope, methodology, and experimental validation of the proposed solution.

### 1. Misleading use of event signals

The paper opens by highlighting the advantages of event cameras, such as high temporal resolution, wide dynamic range, and the absence of motion blur (Lines 15-16). This led me to expect that the paper would leverage event signals to enhance demosaicing. However, the focus is solely on RGB signal demosaicing, and event signals are only considered as a source of missing values. Thus, the introduction gives a misleading impression.

**I suggest the authors revise the introduction to clarify the actual scope of the paper, focusing on the challenges of demosaicing with missing color information due to event pixels, rather than emphasizing the general advantages of event cameras, which are not directly utilized in the method.**

###  2. Mismatch between introduction claims and experimental evidence

In Line 60-61, the authors claim that the HybridEVS design improves low-light imaging and captures high-speed objects effectively. However, none of the experiments provide evidence for low-light imaging or high-speed object capture. All the data used are simulated, and there is no real-world evaluation in challenging conditions. **I suggest the authors either provide experimental results demonstrating these capabilities, such as tests under low-light and high-speed scenarios, or remove these claims if they are not directly relevant to the current work. This would ensure that the experimental results are aligned with the claims in the introduction.**

### 3. Incomplete response to challenges

While the authors correctly outline three major challenges (absence of color information at event pixel locations, joint demosaicing and denoising, and model parameter reduction), the methodology does not fully address all these points. Challenge 1, concerning reconstruction quality due to missing color information, is handled well through the Quad-to-Quad (Q2Q) approach. However, Challenge 2, regarding joint demosaicing and denoising, is not sufficiently addressed. There is no explicit discussion of denoising in the method, and although Line 373 mentions the MIPI dataset containing real-world data with noise, the experiments are solely based on simulated datasets. This makes the claim of addressing noise issues unsubstantiated without a noise model or calibration. **I recommend that the authors provide a more detailed explanation of how their method addresses the denoising aspect, particularly in the Quad-to-Quad stage. They should include experiments with real-world noisy data to validate their claims about joint demosaicing and denoising, ensuring that the solution is robust to real-world conditions.**

### 4. The uniqueness of the TSANet architecture
The proposed TSANet architecture consists of two stages: Quad-to-Quad and Quad-to-RGB, which resemble approaches seen in prior work, such as those in the MIPI Challenge 2024 (e.g., TSIDN). The main distinction claimed by the authors is the use of a state space model. However, it is unclear why the state space model is particularly well-suited for the demosaicing task or how it provides unique advantages over conventional models. The paper would benefit from a more in-depth analysis and comparison with traditional architectures, specifically highlighting the role of the state space model in this context. **Therefore, I suggest the authors provide a more in-depth analysis of why the state space model is particularly suitable for demosaicing. They should compare it to conventional models in terms of performance, especially in tasks specific to HybridEVS, and give examples of how it improves performance.**

### 5. Lack of loss function details

The paper does not provide a detailed explanation of the loss functions used during training. Without this information, it is difficult to assess how the model learns to inpaint and denoise in the Quad-to-Quad stage. A detailed description of the loss terms and their influence on the model's performance would improve the paper’s clarity. **I recommend that the authors include a detailed description of the loss functions used, explaining how each component contributes to the learning process. This should specifically cover the inpainting and denoising tasks in the Quad-to-Quad stage, clarifying how the model optimizes for both tasks.**

### 6. Inadequate real-world data validation

Despite the mention of real-world data in the MIPI dataset (Line 375), it is unclear which images in the experiments come from real-world data. The caption in Figure 7 does not specify these images are real or simulated. Moreover, the claim in Line 480 regarding TSANet's ability to handle high-speed dynamic range scenes is confusing. The displayed images do not appear to represent HDR scenes, and there is no way for the reader to verify if they involve high-speed objects. **I suggest that the authors clearly label which images are from real-world data and which are simulated in all figures. Additionally, they should provide more concrete evidence to support their claims about handling high-speed dynamic range scenes. For example, they could include multiple frames from video sequences or use specific metrics to measure performance in high-speed or HDR content.**

### Conclusion

This paper addresses a timely and relevant problem in the demosaicing of new sensor designs for HybridEVS. However, the introduction and scope need refinement to avoid confusion, and the methodology and experiments should be better aligned with the paper's claims. The authors are encouraged to focus on the core challenges and provide more robust experimental evidence, especially with real-world data and a clearer discussion of denoising and noise modeling. In summary,  I encourage the authors to focus on the **core challenges** and provide more robust experimental evidence, especially using **real-world** data and a clearer discussion of denoising and noise modeling.

**Strengths:**

The paper explores a new and important research topic: demosaicing for Hybrid Event-based Vision Sensors (HybridEVS). This is a valuable area of study given the increasing interest in event vision (MIPI Demosaic 2024).

The proposed TSANet introduces a lightweight network, which has the potential to be applied on mobile devices. However, the authors have not conducted experiments to validate its performance on edge computing (challenges iii).

**Weaknesses:**

Please refer to Summary

**Questions:**

Please refer to Summary

---

### Official Review · Reviewer_diP8 · 2024-10-27

**Soundness:** 3
**Presentation:** 3
**Contribution:** 2
**Rating:** 3
**Confidence:** 4

**Summary:**

This paper proposes a network named TSANet to solve the Quad Bayer HybridEVS demosaicing problem. It is a lightweight two-stage network based on state space augmented cross-attention, which can handle event pixels inpainting and Quad Bayer demosaicing separately, leveraging the benefits of dividing complex tasks into manageable subtasks and learning them through a two-step training strategy to enhance robustness.

**Strengths:**

* The proposed two-stage network structure design seems to be effective in handling such kind of data.
* The proposed method requires less number of parameters, which could be efficient in deploying on limited-resource mobile devices.

**Weaknesses:**

* The motivation of proposing a two-stage network structure design is not that clear.  As shown in Line200, the authors say that "all-in-one models often struggle to extract the inner connection between position and color", but do not provide any explanation or proof. They only let the readers to see the experimental results in Fig.6. It seems that it is more like story-telling (\eg, "in our experiments we found that doing xx could be better than doing yy") instead of giving in-depth analysis on why all-in-one models cannot achieve good performance (\eg, "we observe that xxx, and this is because xxx").

* The compared methods are not that relevant to the Quad Bayer HybridEVS demosaicing problem. Only  DemosaicFormer is designed for such a problem. Please replace the other compared methods with the methods proposed in MIPI 2024 Challenge on Demosaic for Hybridevs Camera [a]. Only if the proposed method outperforms the methods in [a] we can say that the proposed method achieves the state-of-the-art performance.

* The experimental results may be wrong. For example, for DemosaicFormer, they achieves 44.8464 PSNR and 0.9854 SSIM in [a], but in the experimental results the authors report that it only achieves 39.35 PSNR and 0.981 SSIM (on MIPI dataset, which is the same dataset used in [a]). I wonder why the results of DemosaicFormer become worse.

* The visual results are not that good. For example, it seems that the visual results of    Demosaicformer are better than the proposed method in both Fig.5, Fig. 6, and Fig. 7.

* The writing quality is not that good. For example, the conference names in Ref are not unified (\eg, for CVPR, the authors use both "Proceedings of the IEEE/CVF conference on computer vision and pattern recognition", "Proceedings of the IEEE Conference on Computer Vision and Pattern Recognition (CVPR)"; for ECCV, the authors use both "ECCV", "European Conference on Computer Vision", and "European conference on computer vision").

  [a]  MIPI 2024 Challenge on Demosaic for Hybridevs Camera: Methods and Results

**Questions:**

* What is the definition of the term "position information" ($P_e$ and $P_q$) in Line 208-210? I cannot figure out the physical meaning. It seems that the symbols $P_e$ and $P_q$ only exist in Eq. (2).
* What is the loss function used for training? I cannot find it.
* Did the proposed method participate to the MIPI 2024 Challenge on Demosaic for Hybridevs Camera?
* Please add the details about how to make the synthetic datasets.

---

### Official Review · Reviewer_zcWM · 2024-10-31

**Soundness:** 2
**Presentation:** 2
**Contribution:** 2
**Rating:** 3
**Confidence:** 5

**Summary:**

This paper presents TSANet, a novel lightweight two-stage network architecture for Quad Bayer HybridEVS demosaicing. The proposed method addresses the aliasing and artifacts in demosaicing, particularly in mobile and resource-constrained devices. TSANet separates the task into event pixel inpainting & denoising and Quad Bayer demosaicing, using state space augmented blocks (Cross-Swin State Block, Conv State Block, and Residual Vision State Space), and innovative attention mechanisms (Quad Bayer Cross Swin Attention and Spatial Position Attention) to leverage both position and color information for improved image reconstruction. The authors claim that TSANet outperforms the previous state-of-the-art method DemosaicFormer in terms of PSNR and SSIM across seven datasets while reducing parameter and computation costs.

**Strengths:**

S1. The research on lightweight hybrid event camera demosaicing architectures holds significant potential for advancing the field of event cameras. In the experiments, the proposed TSANet-s markedly outperforms the SOTA in terms of performance while maintaining the lowest parameter count and complexity.

S2. Integrating SSM with window attention is an effective approach, as it substantially reduces model complexity while balancing both global and local information.

S3. The authors incorporated the quad Bayer pattern information in the encoding phase, which is a task-specific design for hybrid event camera demosaicing.

**Weaknesses:**

W1: The combination of state-space models with attention does not appear enough novel. Additionally, the effects of the proposed QCSA and SPA as shown in the ablation study are minimal.

W2: The paper lacks an in-depth discussion of integrating the Quad Bayer pattern's positional information.  This aspect should be one of the primary focus.

W3: On Page 4, Line 215, previous studies have shown that pretraining sub-networks can improve performance and inference stability, yet there is no citation to support this claim.

W4: The comparative methods are outdated, and the paper should include comparisons with the latest lightweight restoration methods.   Moreover, the specific variant of TSANet (s, m, or l) used in visual comparisons in Figures 5, 6, and 7 is unclear, which could lead to unfair comparisons.

W5: The ablation studies lack an ablation for the positional information and the FFM.  Additionally, the descriptions in the ablation study are unclear.

W6: There are flaws in the writing and visualization. For instance, there is redundancy in the figures and tables, such as Figure 9 and Table 2 convey the same information. Additional visualizations such as feature visualization in the network are necessary.

**Questions:**

Q1: The authors considered the impact of positional information but only incorporated relevant designs in QCSA and SPA.   Why is no task-specific design introduced in the state-space model, such as implementing a novel scanning mechanism?

Q2: What is the purpose of designing the network as a two-stage architecture? Apart from the two-step training strategy, there seems to be no loss constraint for the Q2Q network, making the significance of the two-stage design unclear. The authors could provide intermediate visualizations from the first-stage network or relevant experiments to support this design choice.

Q3: Please add the experiments mentioned in W4 and W5 and provide a detailed qualitative comparison. Additionally, the writing and visualizations in the manuscript should be improved, as noted in W6.

Overall, if the authors can address my concerns, I would consider raising my score to marginally above the acceptance threshold.

---

### Official Review · Reviewer_aRQW · 2024-11-03

**Soundness:** 2
**Presentation:** 3
**Contribution:** 2
**Rating:** 5
**Confidence:** 3

**Summary:**

This paper proposed a lightweight network for joint demosaicing and denoising of Quad Bayer HybridEVS. The network applies a two-stage framework that firstly inpaints event pixels then convert the Quad Bayer into RGB images. Experimental results demonstrates the effectiveness of the proposed network in different dataset.

**Strengths:**

•	The proposed RVSS module demonstrates effectiveness by significantly reducing model parameters while maintaining competitive performance levels. This efficiency can be particularly advantageous for resource-constrained environments, enabling the deployment of complex vision models on devices with limited computational power. The parameter reduction, achieved without notable sacrifices in accuracy or quality, highlights the RVSS module’s potential for scalability and its suitability for lightweight applications in vision tasks.

**Weaknesses:**

•	The paper's relevance to event-based vision is unclear, as it lacks components specifically tailored for processing event signals. There is no dedicated mechanism or module designed to leverage the unique properties of event-based input. This raises questions about the paper’s contributions to event-based vision specifically.

•	The paper does not detail the loss function or the two-stage training strategy, both of which are crucial for understanding the network’s optimization and performance. Without information on the loss function, it is difficult to assess how the model balances competing objectives or encourages specific qualities in the output. Similarly, the lack of clarity around the two-stage training approach leaves readers uncertain about its purpose, implementation details, and impact on final model performance.

•	The motivation for each of the proposed network modules is insufficiently explained. For instance, while the authors employ an SSM (Spatiotemporal Synthesis Module), it is unclear why this choice would be advantageous over other architectures, such as transformers or CNNs, that are commonly used in similar tasks. A more detailed discussion on the rationale for using SSM and how it aligns with the task requirements would strengthen the paper.

•	In Figure 8, the visual differences in ablation results are difficult to discern. This limits the reader’s ability to evaluate the impact of individual components or modifications effectively.

**Questions:**

•	Could the authors elaborate on how the Fourier transformation maps positional information into high-frequency features? Specifically, how does this process benefit the network in capturing spatial details compared to conventional positional encoding methods?

•	How are the event pixels simulated in the proposed dataset? It would be helpful to understand the assumptions, methodology, and any limitations inherent in this simulation, especially as real event-based sensors exhibit unique properties that might differ from simulated data.

---

### Note · Authors · 2024-11-14

**Comment:**

We sincerely appreciate the reviewers for their insightful comments and constructive suggestions, which greatly helped improve the quality of our work. We also extend our heartfelt thanks to the Area Chair (AC) and Program Chair (PC) for their guidance and support throughout the review process.

**Withdrawal Confirmation:**

I have read and agree with the venue's withdrawal policy on behalf of myself and my co-authors.